# Demographic Considerations in Incenting Reuse of Corrugated Cardboard Boxes

Harshwardhan Ketkale and Steven Simske *

Systems Engineering Department, Colorado State University, Fort Collins, CO 80523, USA;
harshwardhan.ketkale@colostate.edu
* Correspondence: steve.simske@colostate.edu; Tel.: +1-970-241-5692

**Abstract:** Climate change is heavily impacted by greenhouse gases. Many sustainability efforts directly or indirectly affect greenhouse gas (GHG) emissions into the environment. In order to address climate change, sustainability efforts are promoted all around the world. The need to motivate the general population was identified by authors in their previous research. This paper proposes to use a positive reinforcement ethos as a psychological incentive to motivate the general population. This paper further examines the findings of the previous paper to better construct the structure of motivating the general population with the use of this positive reinforcement ethos. This paper attempts to segment the general population based on demographic information including age, gender, awareness of climate change, and current recycling efforts to examine its relevance with persuasion and operant conditions. Further, this paper also tests the hypothesis of using entropy as a tool to identify confusing/leading questions on the survey. Two different sustainability effort options are explored: returning and reusing Corrugated Cardboard Boxes (CCBs). An online survey is conducted, and its data are analyzed to test these hypotheses. The results indicate that reusing CCBs is statistically significantly preferred over returning them. Also, ethos and aesthetics are statistically significantly preferred over logos and pathos. Segmenting the general population based on demographic does not yield any significant effect on motivating the general population. The results of this study can be applied to motivate the general population for different sustainability efforts such as promoting green energy, waste management, and other initiatives.

**Keywords:** sustainability; incentives; motivation; reuse; entropy

## 1. Introduction

The United States Environmental Protection Agency (EPA) defines sustainability as "everything that we need for our survival and well-being depends, either directly or indirectly, on our natural environment. To pursue sustainability is to create and maintain the conditions under which humans and nature can exist in productive harmony to support present and future generations" [1]. Thus, promoting sustainability efforts is important, at a minimum, since humans are directly or indirectly dependent on the environment. According to the United Nations Climate Action (UNCA) [2], the largest contributor to global climate change is the use of fossil fuels and the carbon emissions from it. The seven causes identified by the United Nations Climate Action are generating power, manufacturing goods, cutting down forests, using transportation, producing food, powering buildings, and overconsuming. The recycling process, as seen in the recent literature reviews, is one of the options to reduce greenhouse gas emissions [3–9]. The Intergovernmental Panel on Climate Change (IPCC) states that "Recycling reduces GHG emissions through lower energy demand for production (avoided fossil fuel) and by substitution of recycled feedstocks for virgin materials" [10] (p. 602). Although recycling would help in reducing GHG emissions, the motivation for recycling is lacking in the general population, as observed by Abila [11], Gilli et al. [12], Kattoua et al. [13], Seacat and Boileau [14], and Li et al. [15]. The

authors, in a previous work [16], proposed ways to encourage the general population to reuse Corrugated Cardboard Boxes (CCBs) instead of landfilling them with the use of incentive methods combining operant conditioning and persuasion preferences. The authors, moreover, showed that a lifecycle assessment and economic cost analysis of reusing CCBs is possible [17]. The current research tries to reduce carbon emissions from five of the seven causes (apart from producing food and overconsumption) identified by UNCA in the case of CCBs. Promoting sustainable efforts is important, which is the reason behind focusing on studying the incentive techniques and recommendations from the authors' previous papers in depth [16,17]. The authors [16] concluded that in terms of motivating the general population for sustainable efforts, segmenting the general population into groups and incenting each group according to their preference is ineffective. A more general incentivization approach for the general population was recommended. In order to effectively motivate the general population for sustainable efforts, it is important to evaluate this claim of segmentation using additional segmenting options. While conducting surveys, it is a common practice among researchers to collect demographic data and analyze the overall data based on subcategories. This paper explores additional segmenting options based on demographic data including age, gender, awareness of environment/climate change, and current recycling efforts.

One of the causes mentioned by the UN for climate change is transportation. It is important in terms of the lifecycle of CCBs to evaluate the transportation option for the proposed reuse phase. Thus, it is worth exploring the options in the collection of CCBs for the reuse phase. One approach is to have the general population assign the used CCBs to a specific bin called the "reuse" bin. These CCBs are then collected by a truck and transported to a specific location for further processing. The other option is that individuals gather their used CCBs and personally drive to the nearest specific location (collection site) for drop-off. These two explored options are very different and require different levels of motivation and carbon emissions. The hypothesis here is that more effort is required for individuals to drive to the collection site. Thus, they would need to be more motivated compared to the other option of assigning CCBs to the reuse bins. The carbon emissions vary for both options, as the option where individuals would need to drive to the collection site would have more carbon emissions as more vehicles are used. Thus, the survey attempts to elicit which method of collecting CCBs for recycling (a reuse bin or dumping at a reuse site) is more appealing to the general population, with respect to operant conditions and persuasion techniques.

Many research papers discuss the methodology for developing a questionnaire that avoids the use of leading/confusing questions [18–23]. The authors in [16] also proposed a new tool for using entropy calculations to evaluate the questions asked on the survey to identify if any particular question is biased/confusing/double-barreled. This research further investigates if entropy can be used to identify problems with the questions asked on surveys.

Similar studies where the general population was incentivized to reuse instead of the recycling process were not found in the literature review. Research papers [11,12,15,24] tried to promote sustainable efforts in the general population by using incentives. Based on waste management service charges, the authors of [25–27] tried to incentivize the general population to reduce waste generation. The indirect incentive used by [25–27] was to charge the individual household based on the weight of the waste they wanted to dispose. Gibovic and Bikfalvi [28] studied the use of virtual currency as a means of financial incentive to increase the plastic recycling rate in the general population. Thus, the literature review indicated a need to motivate the general population toward sustainable efforts. Also, a unique method of incentivization like using operant conditioning and persuasion techniques was not found.

Overall, this paper tests the hypothesis that segmenting the general population based on age, gender, recycling efforts, and awareness about environment/climate change has a significant impact on people's preference over incentives. By testing this hypothesis,

the findings may add a new way of motivating the general population to the body of knowledge. This research may also prove the use of entropy to analyze the survey data and examine the survey questions.

## 2. Materials and Methods

Reference [16] concludes that motivating the general population by segmenting them into different groups did not add significant value (measured in terms of overall cost of incenting). In order to further examine this methodology for incentivizing sustainable efforts from the general population, an additional survey was carried out. Survey questions and methodology were reviewed and approved by Colorado State University's Institutional Review Board (IRB). The methodology used in this paper is to conduct the survey and analyze data to test the hypothesis. The research methodology used in this paper is consistent with the previous research [16,17] on which this research paper is based. This research method includes carefully wording the questionnaire to test the required hypothesis as well as making sure that the questions or options are not leading/confusing. In order to avoid these errors, entropy is calculated for all the questions. The survey in [16] also uses entropy calculations to determine the quality/clarity of questions with respect to participants' responses. Entropy is a measure of randomness [16], with random data having higher entropy, and vice versa. It is important to test the hypothesis behind the use of entropy as a unit of measure to evaluate the clarity of questions. The authors identified a few questions from [16] under survey #1, which can be categorized as confusing or double-barreled questions. These questions could be confusing to the participants (indicated by exaggerated entropy, a measure of randomness). Thus, rephrasing the questions for clarity and evaluating entropy change would test the hypothesis.

*Survey*

There were 58 questions in total on this survey. The objective of this survey was to further evaluate and test the results and conclusions from [16] about incentivizing the general population without the necessity of performing market segmentation. This survey evaluates the preferences of the general population with respect to two different ways of collecting processes for the reuse phase (assigning and returning). This survey also evaluates the "entropy" tool by rephrasing the question with high entropy from survey #1 in [16]. Below are the types of questions that were included in this survey.

1. Six questions to note the demographics of the participants participating in this survey.
2. Questions to evaluate the collection process by assigning CCBs to reuse bins.
    a. Multiple-choice questions (12 questions)
3. Questions to evaluate the collection process by returning CCBs to a specific location.
    a. Multiple-choice questions (12 questions)
4. Questions to assess persuasion preferences.
    a. Likert-type questions (20 questions)
5. Questions to evaluate entropy change by rephrasing.
    a. Likert-type questions (5 questions)

This survey evaluates the possibility of adding value in motivation by segmenting the general population with respect to demographics. Additionally, it identifies the general population's preferences over the collection process of CCBs for reusing.

## 3. Results

This survey was published online on the social media platform LinkedIn. The survey was also sent to participants from the survey conducted in [16,17]. Additionally, this survey was distributed to the students, faculty, and staff of Colorado State University. The survey was created, and the responses were collected online using the Qualtrics tool. The survey was active for 50 days and received 151 responses. Responses for the survey were provided

by participants from seven countries on four continents. Qualtrics metadata show that the survey received responses from seven countries. The median time to complete this survey was 9.18 min. Once the responses were collected by the Qualtrics tool, the data were then exported and analyzed in Excel and by the IBM SPSS tool.

### 3.1. Results and Analysis for Assigning Method

3.1.1. Results for Multiple-Choice Questions

Multiple-choice questions were asked with two options representing two persuasion techniques or two operant conditions each for the assigning approach. Thus, the four persuasion techniques (Ethos, Pathos, Logos, and Aesthetics) and four operant conditions (Positive reinforcement, Negative reinforcement, Positive punishment, and Negative punishment) were compared to each other within their respective category. Table 1 gives the results for the multiple-choice questions for assigning CCBs.

**Table 1.** Results for multiple-choice questions for assigning method.

| | | | | | |
|---|---|---|---|---|---|
| Q7 | Ethos<br>Pathos | 130<br>21 | Q19 | Positive Reinforcement<br>Positive Punishment | 85<br>66 |
| Q8 | Ethos<br>Logos | 93<br>58 | Q20 | Positive Reinforcement<br>Negative Punishment | 92<br>59 |
| Q9 | Aesthetics<br>Ethos | 85<br>66 | Q21 | Negative Reinforcement<br>Positive Reinforcement | 85<br>66 |
| Q10 | Pathos<br>Logos | 76<br>75 | Q22 | Positive Punishment<br>Negative Punishment | 87<br>64 |
| Q11 | Aesthetics<br>Pathos | 121<br>30 | Q23 | Negative Reinforcement<br>Positive Punishment | 87<br>64 |
| Q12 | Aesthetics<br>Logos | 108<br>43 | Q24 | Negative Reinforcement<br>Negative Punishment | 109<br>42 |

3.1.2. Analysis of Multiple-Choice Questions

To analyze the answers for the general population's preferences, a chi-square test was conducted to evaluate if one of the two options was significantly preferred by the participants. A chi-square test is used to statistically evaluate the goodness of fit between the expected values and measured values. The total number of participants was 151; thus, the expected value here is the midpoint between 0 and 151, or 75.5. Tables 2 and 3 give the analysis results for assigning CCBs.

**Table 2.** Chi-square analysis results for persuasion techniques for assigning method.

| Question Number | Persuasion Technique | Observed Score | Expected Score | Chi-Square Score | *p*-Value |
|---|---|---|---|---|---|
| Q7 | Ethos<br>Pathos | 130<br>21 | 75.5 | 78.68 | <0.001 * |
| Q8 | Ethos<br>Logos | 93<br>58 | 75.5 | 8.11 | 0.004 * |
| Q9 | Aesthetics<br>Ethos | 85<br>66 | 75.5 | 2.39 | 0.122 |
| Q10 | Pathos<br>Logos | 76<br>75 | 75.5 | 0.01 | 0.935 |
| Q11 | Aesthetics<br>Pathos | 121<br>30 | 75.5 | 54.84 | <0.001 * |
| Q12 | Aesthetics<br>Logos | 108<br>43 | 75.5 | 27.98 | <0.001 * |

An asterisk (*) indicates that the results are statistically significant at $p \leq 0.01$.

**Table 3.** Chi-square analysis results for operant conditioning for assigning method.

| Question Number | Operant Condition | Observed Score | Expected Score | Chi-Square Score | *p*-Value |
|---|---|---|---|---|---|
| Q19 | Positive Reinforcement | 85 | 75.5 | 2.39 | 0.122 |
| | Positive Punishment | 66 | | | |
| Q20 | Positive Reinforcement | 92 | 75.5 | 7.21 | 0.007 * |
| | Negative Punishment | 59 | | | |
| Q21 | Negative Reinforcement | 85 | 75.5 | 2.39 | 0.122 |
| | Positive Reinforcement | 66 | | | |
| Q22 | Positive Punishment | 87 | 75.5 | 3.50 | 0.061 |
| | Negative Punishment | 64 | | | |
| Q23 | Negative Reinforcement | 87 | 75.5 | 3.50 | 0.061 |
| | Positive Punishment | 64 | | | |
| Q24 | Negative Reinforcement | 109 | 75.5 | 29.72 | <0.001 * |
| | Negative Punishment | 42 | | | |

An asterisk (*) indicates that the results are statistically significant at $p \leq 0.01$.

### 3.2. Results and Analysis for Returning Method

#### 3.2.1. Results for Multiple-Choice Questions

Multiple-choice questions were asked with two options representing two persuasion techniques or two operant conditions each for the returning approach. Thus, the four persuasion techniques and four operant conditions were compared to each other within their respective categories. Table 4 gives the results for the multiple-choice questions for returning CCBs.

**Table 4.** Results for multiple-choice questions for returning method.

| Q13 | Ethos | 129 | Q25 | Positive Reinforcement | 93 |
|---|---|---|---|---|---|
| | Pathos | 22 | | Positive Punishment | 58 |
| Q14 | Ethos | 81 | Q26 | Positive Reinforcement | 105 |
| | Logos | 70 | | Negative Punishment | 46 |
| Q15 | Aesthetics | 82 | Q27 | Positive Reinforcement | 79 |
| | Ethos | 69 | | Negative Reinforcement | 72 |
| Q16 | Logos | 86 | Q28 | Positive Punishment | 86 |
| | Pathos | 65 | | Negative Punishment | 65 |
| Q17 | Aesthetics | 123 | Q29 | Negative Reinforcement | 82 |
| | Pathos | 28 | | Positive Punishment | 69 |
| Q18 | Aesthetics | 90 | Q30 | Negative Reinforcement | 105 |
| | Logos | 61 | | Negative Punishment | 46 |

#### 3.2.2. Analysis for Multiple-Choice Questions

A chi-square test was again conducted to evaluate if one of the two options is significantly preferred by the participants. The expected value here is considered to be 75.5, as mentioned earlier. Tables 5 and 6 give the analysis results for returning CCBs.

### 3.3. Results and Analysis for Likert Scale Questions

#### 3.3.1. Results for Likert Scale Questions

Likert scale questions were asked to evaluate the general population's preferences for persuasion techniques. Likert scale questions include five options as follows: strongly agree, somewhat agree, neither agree nor disagree, somewhat disagree, and strongly disagree. To evaluate the results based on the responses, a linear scoring scale was considered with strongly disagree as 1 and strongly agree as 5. Table 7 gives the results for the Likert scale questions.

**Table 5.** Chi-square analysis results for persuasion techniques for returning method.

| Question Number | Persuasion Technique | Observed Score | Expected Score | Chi-Square Score | *p*-Value |
|---|---|---|---|---|---|
| Q13 | Ethos<br>Pathos | 129<br>22 | 75.5 | 75.82 | <0.001 * |
| Q14 | Ethos<br>Logos | 81<br>70 | 75.5 | 0.80 | 0.370 |
| Q15 | Aesthetics<br>Ethos | 82<br>69 | 75.5 | 1.11 | 0.290 |
| Q16 | Logos<br>Pathos | 86<br>65 | 75.5 | 2.92 | 0.087 |
| Q17 | Aesthetics<br>Pathos | 123<br>28 | 75.5 | 59.76 | <0.001 * |
| Q18 | Aesthetics<br>Logos | 90<br>61 | 75.5 | 5.57 | 0.018 |

An asterisk (*) indicates that the results are statistically significant at $p \leq 0.01$.

**Table 6.** Chi-square analysis results for operant conditioning for assigning approach.

| Question Number | Operant Condition | Observed Score | Expected Score | Chi-Square Score | *p*-Value |
|---|---|---|---|---|---|
| Q25 | Positive Reinforcement<br>Positive Punishment | 93<br>58 | 75.5 | 8.113 | 0.004 * |
| Q26 | Positive Reinforcement<br>Negative Punishment | 105<br>46 | 75.5 | 23.053 | <0.001 * |
| Q27 | Positive Reinforcement<br>Negative Reinforcement | 79<br>72 | 75.5 | 0.325 | 0.568 |
| Q28 | Positive Punishment<br>Negative Punishment | 86<br>65 | 75.5 | 2.921 | 0.087 |
| Q29 | Negative Reinforcement<br>Positive Punishment | 82<br>69 | 75.5 | 1.119 | 0.290 |
| Q30 | Negative Reinforcement<br>Negative Punishment | 105<br>46 | 75.5 | 23.053 | <0.001 * |

An asterisk (*) indicates that the results are statistically significant at $p \leq 0.01$.

### 3.3.2. Analysis of Likert Scale Questions

To analyze the data from Likert scale questions, an independent *t*-test was calculated to compare each pair of persuasion technique scores. Table 8 gives the results of the independent *t*-tests on the Likert scale questions.

**Table 7.** Results for Likert scale questions.

| Persuasion Technique | Question Number | Score | Mean Score |
|---|---|---|---|
| Aesthetics | Q35<br>Q40<br>Q45<br>Q50<br>Q54 | 4.37<br>4.31<br>4.32<br>4.28<br>4.13 | 4.28 |
| Ethos | Q32<br>Q36<br>Q41<br>Q46<br>Q51 | 4.19<br>4.25<br>4.28<br>4.07<br>4.26 | 4.21 |
| Logos | Q34<br>Q39<br>Q44<br>Q48<br>Q53 | 4.06<br>4.06<br>3.80<br>2.21<br>3.15 | 3.46 |

**Table 7.** *Cont.*

| Persuasion Technique | Question Number | Score | Mean Score |
|---|---|---|---|
| Pathos | Q33 | 3.56 | |
| | Q38 | 4.04 | |
| | Q42 | 3.99 | 3.98 |
| | Q47 | 4.05 | |
| | Q52 | 4.28 | |

**Table 8.** Independent *t*-test results of Likert scale questions.

| Comparison of Persuasion Techniques | *t*-Value | *p*-Value |
|---|---|---|
| Ethos (4.21) with Pathos (3.98) | 2.25 | 0.024 * |
| Ethos (4.21) with Logos (3.46) | 2.59 | 0.013 * |
| Aesthetics (4.28) with Ethos (4.21) | 1.57 | 0.073 |
| Aesthetics (4.28) with Pathos (3.98) | 2.94 | 0.007 * |
| Aesthetics (4.28) with Logos (3.46) | 2.84 | 0.008 * |
| Logos (3.46) with Pathos (3.98) | 1.61 | 0.069 |

An asterisk (*) indicates that the results are statistically significant at $p \leq 0.05$.

### 3.4. Results and Analysis of Data Based on Demographics

3.4.1. Results Based on Demographics

In total, six demographic questions were asked. These questions help to identify a participant's age, gender, awareness of climate change, and current recycling efforts. Figure 1 shows the results of the distribution of participants based on the respective demographic information.

3.4.2. Analysis of Data Based on Demographics

The data are partitioned by demographics and analyzed based on the question types. The detailed results of the analyzed data are given in Appendix A. The sections below give a brief summary of those results.

Summary of Analyzed Data from Multiple-Choice Questions

Table 9 gives a summary of the results for the multiple-choice questions. Additionally, a chi-square test was conducted to analyze the data.

**Table 9.** Summary of results for chi-square test on multiple-choice questions.

| | | | Ethos | Pathos | Logos | Aesthetics | Positive Reinforcement | Positive Punishment | Negative Punishment | Negative Reinforcement |
|---|---|---|---|---|---|---|---|---|---|---|
| Based on Age | 18–30 | Mean | 30.3 | 10.7 | 29.8 | 36.0 | 30.2 | 32.8 | 17.0 | 26.0 |
| | | Std. Dev. | 12.5 | 4.9 | 5.7 | 9.0 | 6.5 | 6.8 | 4.6 | 6.7 |
| | 31–45 | Mean | 35.3 | 16.3 | 21.7 | 35.8 | 30.2 | 28.3 | 18.8 | 32.7 |
| | | Std. Dev. | 8.9 | 10.2 | 5.2 | 8.2 | 5.3 | 4.6 | 2.4 | 4.9 |
| | 46+ | Mean | 25.7 | 12.0 | 12.3 | 24.8 | 23.2 | 9.8 | 15.8 | 27.2 |
| | | Std. Dev. | 6.2 | 7.7 | 4.6 | 5.8 | 5.0 | 2.3 | 8.8 | 5.4 |
| | Prefer Not to Answer | Mean | 3.3 | 1.3 | 1.7 | 3.6 | 3.2 | 0.7 | 2.0 | 4.2 |
| | | Std. Dev. | 1.4 | 1.4 | 0.5 | 0.5 | 1.0 | 0.5 | 1.7 | 1.0 |
| Based on Gender | Male | Mean | 44.7 | 19.8 | 26.5 | 45.8 | 38.7 | 28.7 | 28.0 | 42.7 |
| | | Std. Dev. | 12.1 | 13.8 | 5.8 | 10.3 | 5.4 | 4.0 | 7.1 | 6.0 |
| | Female | Mean | 46.5 | 19.5 | 37.3 | 50.6 | 44.8 | 41.3 | 23.3 | 44.5 |
| | | Std. Dev. | 15.3 | 8.8 | 9.2 | 9.1 | 9.3 | 8.4 | 3.2 | 8.5 |
| | Prefer Not to Mention | Mean | 2.0 | 0.7 | 1.0 | 2.4 | 2.5 | 0.0 | 1.3 | 2.2 |
| | | Std. Dev. | 0.9 | 1.0 | 0.0 | 0.5 | 0.5 | 0.0 | 1.4 | 1.0 |
| | Non-Binary | Mean | 1.5 | 0.3 | 0.7 | 1.4 | 0.7 | 1.7 | 1.0 | 0.7 |
| | | Std. Dev. | 0.5 | 0.5 | 0.5 | 0.5 | 0.8 | 0.5 | 0.6 | 0.5 |

**Table 9.** *Cont.*

| | | | Ethos | Pathos | Logos | Aesthetics | Positive Reinforce-ment | Positive Punishment | Negative Punishment | Negative Reinforcement |
|---|---|---|---|---|---|---|---|---|---|---|
| Based on Awareness | Tremendous | Mean | 14.7 | 6.0 | 11.5 | 15.6 | 12.2 | 10.5 | 7.8 | 17.5 |
| | | Std. Dev. | 4.1 | 2.4 | 3.4 | 3.5 | 3.7 | 2.1 | 3.9 | 3.0 |
| | High | Mean | 48.7 | 19.3 | 30.3 | 48.8 | 44.0 | 33.3 | 28.2 | 42.5 |
| | | Std. Dev. | 14.7 | 13.7 | 7.3 | 11.0 | 5.3 | 5.6 | 5.8 | 7.1 |
| | Moderate | Mean | 28.7 | 13.0 | 22.3 | 34.0 | 29.0 | 25.5 | 16.0 | 27.5 |
| | | Std. Dev. | 10.3 | 7.6 | 5.0 | 6.1 | 4.9 | 5.3 | 1.9 | 4.7 |
| | Little | Mean | 2.7 | 2.0 | 1.3 | 1.8 | 1.5 | 2.3 | 1.7 | 2.5 |
| | | Std. Dev. | 0.5 | 0.6 | 0.5 | 0.4 | 0.8 | 0.5 | 0.8 | 0.5 |
| Based on Recycling Efforts | Tremendous | Mean | 4.2 | 2.3 | 3.7 | 3.8 | 4.7 | 1.8 | 2.5 | 5.0 |
| | | Std. Dev. | 0.8 | 0.5 | 0.5 | 1.1 | 1.0 | 1.0 | 1.6 | 1.7 |
| | High | Mean | 44.2 | 17.0 | 31.3 | 48.8 | 41.0 | 32.5 | 24.3 | 44.2 |
| | | Std. Dev. | 15.3 | 10.7 | 9.9 | 9.1 | 6.0 | 5.2 | 7.6 | 9.0 |
| | Moderate | Mean | 37.7 | 15.8 | 22.3 | 35.6 | 30.2 | 29.8 | 19.7 | 32.3 |
| | | Std. Dev. | 9.5 | 11.5 | 3.2 | 9.4 | 6.0 | 5.6 | 1.8 | 4.6 |
| | Little | Mean | 7.0 | 3.2 | 7.2 | 10.8 | 8.5 | 6.8 | 6.5 | 6.2 |
| | | Std. Dev. | 4.0 | 1.5 | 1.8 | 2.2 | 1.8 | 0.8 | 2.4 | 1.2 |
| | Very Little | Mean | 1.7 | 2.0 | 1.0 | 1.2 | 2.3 | 0.7 | 0.7 | 2.3 |
| | | Std. Dev. | 0.5 | 0.0 | 0.0 | 0.4 | 0.5 | 0.8 | 0.8 | 1.0 |

Summary of Analyzed Data from Likert Scale Questions

Table 10 below shows the results of the Likert scale questions for persuasion techniques based on the demographics. To analyze the following data, *t*-tests were conducted by comparing the persuasion techniques to each other.

**Table 10.** Independent *t*-test results.

| | | | Aesthetics | Ethos | Logos | Pathos |
|---|---|---|---|---|---|---|
| Based on Age | | 18–30 | 4.40 | 4.40 | 3.63 | 4.06 |
| | | 31–45 | 4.31 | 4.25 | 3.43 | 4.05 |
| | | 46+ | 4.15 | 3.97 | 3.27 | 3.87 |
| | | Prefer Not to Answer | 3.84 | 3.56 | 3.32 | 3.32 |
| Based on Gender | | Male | 4.27 | 4.14 | 3.47 | 3.94 |
| | | Female | 4.32 | 4.30 | 3.48 | 4.05 |
| | | Non-Binary | 4.30 | 4.30 | 2.90 | 3.90 |
| | | Prefer Not to Mention | 3.80 | 3.67 | 3.13 | 3.47 |
| Based on Awareness | | Tremendous | 4.44 | 4.52 | 3.28 | 4.08 |
| | | High | 4.29 | 4.22 | 3.50 | 4.05 |
| | | Moderate | 4.16 | 4.02 | 3.47 | 3.83 |
| | | Little | 4.75 | 4.50 | 3.55 | 4.00 |
| Based on Recycling Efforts | | Tremendous | 4.00 | 3.69 | 2.91 | 3.74 |
| | | High | 4.44 | 4.42 | 3.47 | 4.21 |
| | | Moderate | 4.17 | 4.03 | 3.54 | 3.83 |
| | | Little | 3.97 | 3.97 | 3.43 | 3.43 |
| | | Very Little | 4.73 | 5.00 | 3.00 | 4.87 |

*3.5. Results and Analysis for Entropy Calculations*

3.5.1. Results for Entropy Calculation Questions

In total, five questions were asked on the survey to examine the entropy change. These questions (originally from [16]) were reworded for clarity. Table 11 shows the answers to five Likert scale questions from the survey (originally reworded from [16]).

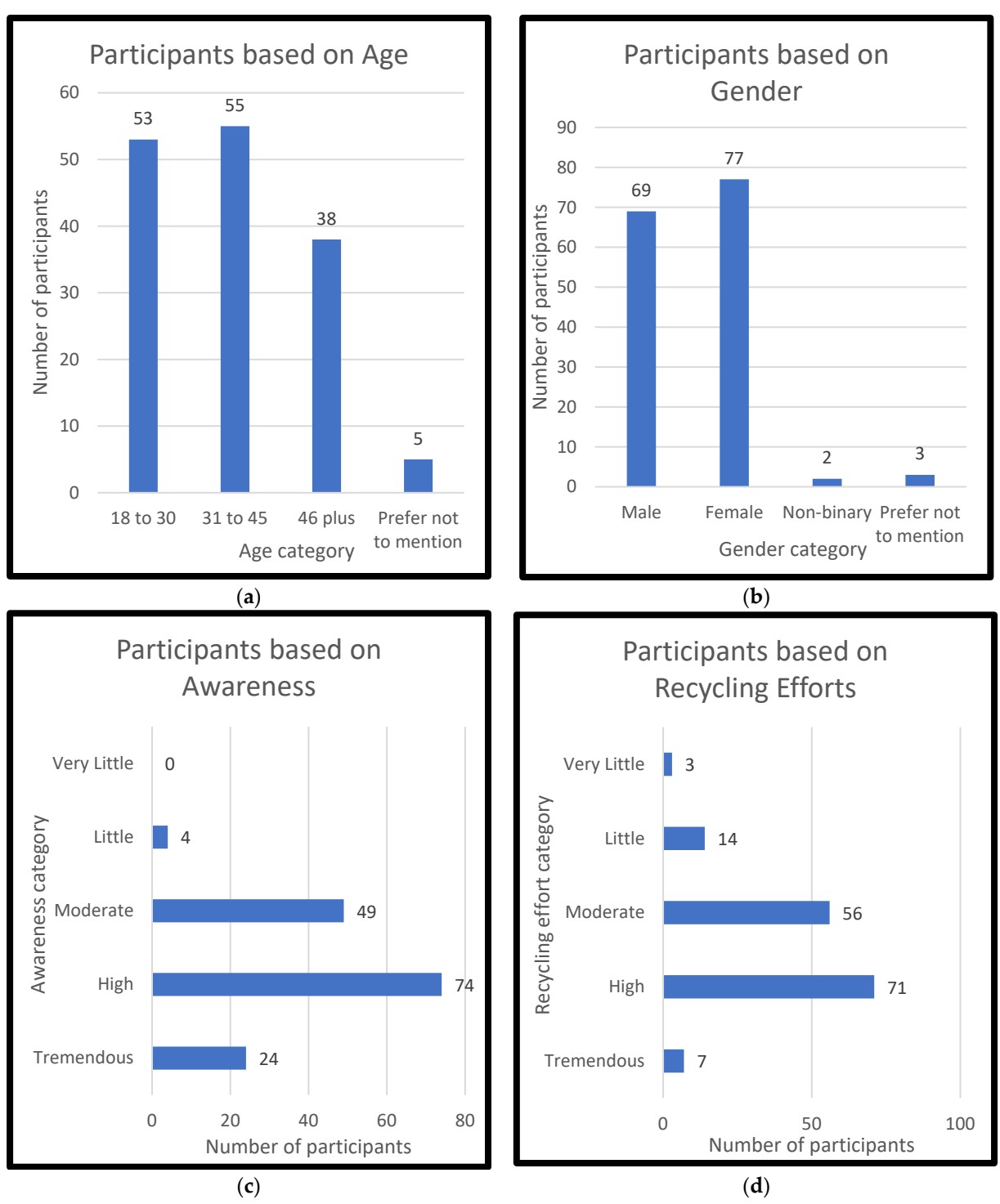

**Figure 1.** Results for participant's demographics based on (**a**) age, (**b**) gender, (**c**) awareness, and (**d**) recycling efforts.

### 3.5.2. Analysis for Entropy Calculations

As explained in [16], entropy is a measure of randomness. Entropy increases as randomness in data increases, and vice versa. In the case of Likert scale questions, high entropy may indicate confusion in questions, as it is primarily expected that the population

of response would be around two main options (strongly disagree or strongly agree). Entropy is calculated by using Equation (1).

$$e = -\sum_{i=1}^{N} p_i \ln(p_i) \tag{1}$$

The entropy values from [16] and this survey are compared, and the differences between the two values are calculated as shown in Table 12.

**Table 11.** Results for Likert scale question for entropy calculation.

| [16] Reference Question Number | Question Number (Current Survey) | Score |
|:---:|:---:|:---:|
| Q13 | Q55 | 3.66 |
| Q14 | Q56 | 4.54 |
| Q17 | Q57 | 3.60 |
| Q27 | Q58 | 3.46 |
| Q32 | Q59 | 4.14 |

**Table 12.** Entropy calculations for reworded questions.

| [16] Reference Question Number | Entropy Values from [16] | Question Number (Current Survey) | Entropy Values from this Survey | Entropy Difference | Entropy Difference (%) |
|:---:|:---:|:---:|:---:|:---:|:---:|
| Q13 | 2.11 | Q55 | 2.10 | 0.01 | 0.47% |
| Q14 | 1.92 | Q56 | 1.30 | 0.62 | 32.29% |
| Q17 | 2.03 | Q57 | 2.11 | −0.08 | −3.94% |
| Q27 | 2.21 | Q58 | 2.11 | 0.10 | 4.52% |
| Q32 | 2.16 | Q59 | 1.74 | 0.42 | 19.44% |

## 4. Discussion

In total, 24 multiple-choice questions were asked on the survey, each of which compared two options among the four choices (for motivation or the operant condition). A chi-square test was carried out to evaluate the participant's preference among the six total comparisons (1 vs. 2, 1 vs. 3, 1 vs. 4, 2 vs. 3, 2 vs. 4, and 3 vs. 4). Tables 1 and 4 give the preference results for assigning CCBs to the reuse bin and returning CCBs to specific locations, respectively. For assigning CCBs with respect to persuasion techniques, it can be observed from Table 2 that the general population statistically significantly more willingly responds toward ethos and aesthetics over logos and pathos. It can be observed that the difference between ethos and aesthetics, as well as the difference between pathos and logos, is not statistically significant. For assigning CCBs with respect to operant conditions, it can be observed from Table 3 that positive reinforcement is statistically significantly preferred over negative punishment, and negative reinforcement is statistically significantly preferred over negative punishment. The other four comparisons of operant conditions to each other are not statistically significantly different. For returning CCBs with respect to persuasion techniques, it can be observed from Table 5 that the general population statistically significantly prefers ethos over pathos and aesthetics over pathos. The difference between logos and both aesthetics and pathos is not statistically significant, whereas the difference between ethos and both logos and aesthetics is not statistically significant. For returning CCBs with respect to operant conditions, it can be observed from Table 6 that positive reinforcement is statistically significantly preferred over both negative punishment and positive punishment. Also, negative reinforcement is statistically significantly preferred over negative punishment. The difference between positive punishment and both negative punishment and negative reinforcement is not statistically significant. Additionally, the difference between positive punishment and negative reinforcement is not statistically significant. Table 7 shows the results for Likert scale questions that evaluate the persuasion preferences of participants. The questions are scored on a scale of 1 to 5, where 1 is strongly

disagree (not preferred) and 5 is strongly agree (preferred). Aesthetics scored highest, followed by ethos with a small, statistically insignificant, margin (0.07), pathos, and logos. The *t*-test shows that at $p \leq 0.05$, ethos and aesthetics are statistically significantly different from pathos and logos. It also shows that the difference between ethos and aesthetics as well as logos and pathos is not statistically significant.

This paper mainly evaluates if segmenting the general population based on their demographic information is an effective approach for motivating the general population to adopt desired sustainable efforts. Table 9 gives the mean and standard deviation of the scores that a multiple-choice question scored with respect to segmenting the general population based on demographics. Tables A1–A4 from Appendix A show the chi-square scores as well as the *p*-values for these multiple-choice questions based on age, gender, awareness, and recycling efforts, respectively. In total, 12 questions (from Q7 to Q18) compare four persuasion techniques to each other. Another 12 questions (from Q19 to Q30) compare four operant conditions to each other. Thus, every pair of persuasion techniques and operant conditions are evaluated twice. Table 10 gives the Likert scale for each persuasion technique with respect to demographics. These scores are calculated by taking the mean of the five questions asked for each persuasion technique. In order to analyze these scores, a *t*-test is conducted by comparing each persuasion technique to the others. Tables A5–A12 from Appendix A show the *t*-value as well as the *p*-value for these Likert scale questions based on age, gender, awareness, and recycling efforts, respectively. In order to better interpret the tabulated results in Appendix A, *p*-values below 0.01 are highlighted in green. Table 11 shows the results for entropy calculations as well as the reference questions from [16] that were reworded. It can be observed from Table 12 that entropy decreased for Q55, Q56, Q58, and Q59. Only Q57 had an increase in entropy by 3.94%. Overall, for five questions, the entropy decreased by 10.56%. The proposed incentivization tool can be used globally, as the overall recycling rate is low compared to other end-of-lifecycle processes. An example of this trend can be observed with the global end-of-lifecycle process of plastic waste. The Organization for Economic Cooperation and Development (OECD) [29,30] shows that as of 2015, 14–18% of global plastic waste is collected for recycling, and 24% of the global plastic waste is thermally treated. The remaining 58–62% of plastic waste ends up in a controlled or uncontrolled landfill. Plastic recycling percentages based on countries [29] include the USA (9%), Australia (12%), Japan (23%), and the EU (30%). As observed in the above data, different countries have different recycling rates for plastics. The plastic recycling example gives a rough idea about the infrastructure in place as well as the difference in the level of motivation for recycling.

From the results and analysis of the multiple-choice questions, it can be observed that for assigning CCBs to the reuse bin, the general population statistically significantly preferred aesthetics and ethos over pathos and logos. This indicates that both ethos and aesthetics persuasion techniques are preferred by the general population for assigning CCBs. In the case of assigning CCBs with respect to operant conditions, no statistically significant preference was found. In the case of returning CCBs with the help of persuasion techniques and operant conditions, no statistically significant preference for a single persuasion technique over another or a single operant condition over another was found. In the case of the Likert scale questions, the results are similar to those of assigning CCBs to reuse bins, with the general population statistically significantly preferring both ethos and aesthetics over logos and pathos. This implies that both aesthetics and ethos are recommended to use to motivate the general population for sustainable efforts. This survey segments the general population based on gender, age, awareness of environment/climate change, and current recycling efforts. The authors conducted *t*-test and chi-square tests on the results and evaluated each sub-category for assigning/returning CCBs with respect to persuasion techniques and operant conditions. It can be concluded that no statistically significant trend in the preferences was observed, implying that the same motivation techniques are broadly applicable across demographics. This paper also examines the use of entropy to evaluate questions for confusion and/or for being double-barreled. The results for the five

reworded questions from [16] show that the entropy decreased by 10.6% overall. As these five questions were identified by the authors to be confusing and double-barreled in [16], they were reworded to make them clearer and more direct.

## 5. Conclusions

The purpose of this paper was to further examine the results from [16] regarding segmenting the general population to effectively motivate it for sustainable efforts. It can be concluded that the segmentation of the general population based on demographics does not yield an effective way of incentivizing the general population for sustainable efforts. Also, to motivate the general population to conduct sustainable efforts, ethos and aesthetics are preferred among the four types of motivation that were evaluated. This supports the claim from [16] about not segmenting the general population for motivation as well as using ethos to motivate the general population. In terms of assigning CCBs to the reuse bin and returning CCBs to a specific location, it can be concluded that assigning CCBs to the reuse bin is preferred by the general population over returning them, which is considered in the life cycle analysis (LCA) for reusing CCBs in [17]. It can be also concluded that entropy may be used in some cases to evaluate the clarity/quality of the survey questions.

Overall, the proposed model from [16,17] and this paper can be tailored to different products and their unique lifecycles. The life cycle analysis (LCA) conducted in [17] would have a different set of data and different processes with respect to the different countries but can still follow the same process. Thus, the overall research provides a repurposable model that can be adjusted for any other products or processes to promote sustainable efforts among the general population and estimate the carbon emissions savings from the LCA. One of the outlooks of this research is the potential application of this new incentive tool of operant condition and persuasion techniques being used to promote sustainable cars, renewable energy, healthcare applications like vaccinations, etc.

The future scope and prospects of this study include identifying a way to convey the incentive message as well as exploring different incentive delivery methods. Research in the area of the implementation of these incentives may play a vital role in further validating this new approach. As seen from the entropy calculations, it is important to frame a clear incentive message. The use of the entropy concept as a tool to evaluate questionnaires may help future researchers to evaluate their questions and improve them accordingly.

**Author Contributions:** Conceptualization, H.K. and S.S.; writing—original draft preparation, H.K.; writing—review and editing, S.S.; visualization, H.K.; supervision, S.S.; funding acquisition, S.S. All authors have read and agreed to the published version of the manuscript.

**Funding:** This research was funded by Colorado State University SoGES (School of Global Environmental Sustainability), Global Challenges Research Team 4 Grant, and APC was funded by Colorado State University's Systems Engineering Department.

**Institutional Review Board Statement:** The study was conducted in accordance with the Declaration of Helsinki and was approved by the Institutional Review Board of Colorado State University (protocol code: 3265; date of approval: 11 March 2022).

**Informed Consent Statement:** Informed consent was obtained from all subjects involved in the study.

**Data Availability Statement:** The (anonymized) data are available upon request and approval by the Colorado State University Institutional Review Board (CSU IRB).

**Acknowledgments:** We wish to show our appreciation to Angie Chromiak for guiding us through the IRB application. We would also like to thank the following people for helping us with the distribution of surveys and administrative tasks: Ingrid, Bridge, Chrissy Charny, Debra Dandaneau, and Mary Gomez.

**Conflicts of Interest:** The authors declare no conflict of interest. The funders had no role in the design of the study; in the collection, analyses, or interpretation of the data; in the writing of the manuscript; or in the decision to publish the results.

## Appendix A

**Table A1.** Analysis of multiple-choice questions based on age. (Green highlight indicates that the results are statistically significant at $p \leq 0.05$).

| | | 18–30 | | | 31–45 | | | 46+ | | | Prefer Not to Mention | | |
|---|---|---|---|---|---|---|---|---|---|---|---|---|---|
| | | Score | Chi^2 Value | p-Value | Score | Chi^2 Value | p-Value | Score | Chi^2 Value | p-Value | Score | Chi^2 Value | p-Value |
| Q7 | Ethos | 46 | 28.69 | <0.001 | 46 | 24.89 | <0.001 | 33 | 20.63 | <0.001 | 5 | 5.00 | 0.025 |
| | Pathos | 7 | | | 9 | | | 5 | | | 0 | | |
| Q8 | Ethos | 28 | 0.17 | 0.68 | 36 | 5.26 | 0.021 | 26 | 5.16 | 0.023 | 3 | 0.20 | 0.654 |
| | Logos | 25 | | | 19 | | | 12 | | | 2 | | |
| Q9 | Ethos | 20 | 3.19 | 0.074 | 24 | 0.89 | 0.345 | 20 | 0.11 | 0.745 | 2 | 0.20 | 0.654 |
| | Aesthetics | 33 | | | 31 | | | 18 | | | 3 | | |
| Q10 | Pathos | 17 | 6.81 | 0.009 | 33 | 2.20 | 0.138 | 23 | 1.68 | 0.194 | 3 | 0.20 | 0.654 |
| | Logos | 36 | | | 22 | | | 15 | | | 2 | | |
| Q11 | Pathos | 8 | 25.83 | <0.001 | 12 | 17.47 | 0.001 | 9 | 10.53 | 0.001 | 1 | 1.80 | 0.179 |
| | Aesthetics | 45 | | | 43 | | | 29 | | | 4 | | |
| Q12 | Logos | 22 | 1.53 | 0.216 | 14 | 13.26 | 0.001 | 6 | 17.79 | 0.001 | 1 | 1.80 | 0.179 |
| | Aesthetics | 31 | | | 41 | | | 32 | | | 4 | | |
| Q13 | Ethos | 46 | 28.69 | <0.001 | 45 | 22.27 | <0.001 | 33 | 20.63 | <0.001 | 5 | 5.00 | 0.025 |
| | Pathos | 7 | | | 10 | | | 5 | | | 0 | | |
| Q14 | Ethos | 22 | 1.53 | 0.216 | 33 | 2.20 | 0.138 | 23 | 1.68 | 0.194 | 3 | 0.20 | 0.654 |
| | Logos | 31 | | | 22 | | | 15 | | | 2 | | |
| Q15 | Ethos | 20 | 3.19 | 0.007 | 28 | 0.02 | 0.892 | 19 | 0.00 | 1 | 2 | 0.20 | 0.654 |
| | Aesthetics | 33 | | | 27 | | | 19 | | | 3 | | |
| Q16 | Pathos | 17 | 6.81 | 0.009 | 25 | 0.46 | 0.500 | 20 | 0.11 | 0.745 | 3 | 0.20 | 0.654 |
| | Logos | 36 | | | 30 | | | 18 | | | 2 | | |
| Q17 | Pathos | 8 | 25.83 | <0.001 | 9 | 24.89 | <0.001 | 10 | 8.53 | 0.003 | 1 | 1.80 | 0.179 |
| | Aesthetics | 45 | | | 46 | | | 28 | | | 4 | | |
| Q18 | Logos | 29 | 0.47 | 0.492 | 23 | 1.47 | 0.224 | 8 | 12.74 | 0.001 | 1 | 1.80 | 0.179 |
| | Aesthetics | 24 | | | 32 | | | 30 | | | 4 | | |
| Q19 | Positive Reinforcement | 23 | 0.93 | 0.336 | 30 | 0.46 | 0.500 | 28 | 8.53 | 0.003 | 4 | 1.80 | 0.179 |
| | Positive Punishment | 30 | | | 25 | | | 10 | | | 1 | | |
| Q20 | Positive Reinforcement | 30 | 0.93 | 0.336 | 36 | 5.26 | 0.021 | 23 | 1.68 | 0.194 | 3 | 0.20 | 0.654 |
| | Negative Punishment | 23 | | | 19 | | | 15 | | | 2 | | |
| Q21 | Positive Reinforcement | 24 | 0.47 | 0.492 | 22 | 2.20 | 0.138 | 18 | 0.11 | 0.745 | 2 | 0.20 | 0.654 |
| | Negative Reinforcement | 29 | | | 33 | | | 20 | | | 3 | | |
| Q22 | Positive Punishment | 40 | 13.76 | 0.001 | 36 | 5.26 | 0.021 | 10 | 8.53 | 0.003 | 1 | 1.80 | 0.179 |
| | Negative Punishment | 13 | | | 19 | | | 28 | | | 4 | | |
| Q23 | Positive Punishment | 30 | 0.93 | 0.336 | 25 | 0.46 | 0.500 | 9 | 10.53 | 0.001 | 0 | 5.00 | 0.025 |
| | Negative Reinforcement | 23 | | | 30 | | | 29 | | | 5 | | |
| Q24 | Negative Punishment | 19 | 4.25 | 0.039 | 17 | 8.02 | 0.004 | 5 | 20.63 | <0.001 | 1 | 1.80 | 0.179 |
| | Negative Reinforcement | 34 | | | 38 | | | 33 | | | 4 | | |
| Q25 | Positive Reinforcement | 31 | 1.53 | 0.216 | 29 | 0.16 | 0.685 | 29 | 10.53 | 0.001 | 4 | 1.80 | 0.179 |
| | Positive Punishment | 22 | | | 26 | | | 9 | | | 1 | | |
| Q26 | Positive Reinforcement | 41 | 15.87 | 0.001 | 36 | 5.26 | 0.021 | 24 | 2.63 | 0.104 | 4 | 1.80 | 0.179 |
| | Negative Punishment | 12 | | | 19 | | | 14 | | | 1 | | |
| Q27 | Positive Reinforcement | 32 | 2.28 | 0.130 | 28 | 0.02 | 0.892 | 17 | 0.42 | 0.516 | 2 | 0.20 | 0.654 |
| | Negative Reinforcement | 21 | | | 27 | | | 21 | | | 3 | | |
| Q28 | Positive Punishment | 39 | 11.79 | 0.001 | 32 | 1.47 | 0.224 | 14 | 2.63 | 0.104 | 1 | 1.80 | 0.179 |
| | Negative Punishment | 14 | | | 23 | | | 24 | | | 4 | | |
| Q29 | Positive Punishment | 36 | 6.81 | 0.009 | 26 | 0.16 | 0.685 | 7 | 15.16 | 0.001 | 0 | 5.00 | 0.025 |
| | Negative Reinforcement | 17 | | | 29 | | | 31 | | | 5 | | |
| Q30 | Negative Punishment | 21 | 2.28 | 0.130 | 16 | 9.62 | 0.001 | 9 | 10.53 | 0.001 | 0 | 5.00 | 0.025 |
| | Negative Reinforcement | 32 | | | 39 | | | 29 | | | 5 | | |

**Table A2.** Analysis of multiple-choice questions based on gender. (Green highlight indicates that the results are statistically significant at $p \leq 0.05$).

| | | Male | | | Female | | | Prefer Not to Mention | | | Non-Binary | | |
|---|---|---|---|---|---|---|---|---|---|---|---|---|---|
| | | Score | Chi^2 Value | p-Value | Score | Chi^2 Value | p-Value | Score | Chi^2 Value | p-Value | Score | Chi^2 Value | p-Value |
| Q7 | Ethos | 60 | 37.70 | <0.001 | 65 | 369.48 | <0.001 | 3 | 3.00 | 0.083 | 2 | 2.00 | 0.157 |
| | Pathos | 9 | | | 12 | | | 0 | | | 0 | | |
| Q8 | Ethos | 42 | 3.26 | 0.070 | 47 | 3.75 | 0.052 | 2 | 0.33 | 0.563 | 2 | 2.00 | 0.157 |
| | Logos | 27 | | | 30 | | | 1 | | | 0 | | |
| Q9 | Ethos | 32 | 0.36 | 0.547 | 32 | 2.20 | 0.138 | 1 | 0.33 | 0.563 | 1 | 0.00 | 1 |
| | Aesthetics | 37 | | | 45 | | | 2 | | | 1 | | |
| Q10 | Pathos | 39 | 1.17 | 0.278 | 34 | 1.05 | 0.305 | 2 | 0.33 | 0.563 | 1 | 0.00 | 1 |
| | Logos | 30 | | | 43 | | | 1 | | | 1 | | |

**Table A2.** *Cont.*

| | | Male | | | Female | | | Prefer Not to Mention | | | Non-Binary | | |
|---|---|---|---|---|---|---|---|---|---|---|---|---|---|
| | | Score | Chi^2 Value | *p*-Value | Score | Chi^2 Value | *p*-Value | Score | Chi^2 Value | *p*-Value | Score | Chi^2 Value | *p*-Value |
| Q11 | Pathos | 12 | 29.35 | <0.001 | 18 | 21.83 | <0.001 | 0 | 3.00 | 0.083 | 0 | 2.00 | 0.157 |
| | Aesthetics | 57 | | | 59 | | | 3 | | | 2 | | |
| Q12 | Logos | 16 | 19.84 | <0.001 | 26 | 8.12 | 0.004 | 1 | 0.33 | 0.563 | 0 | 2.00 | 0.157 |
| | Aesthetics | 53 | | | 51 | | | 2 | | | 2 | | |
| Q13 | Ethos | 59 | 34.80 | <0.001 | 65 | 369.48 | <0.001 | 3 | 3.00 | 0.083 | 2 | 2.00 | 0.157 |
| | Pathos | 10 | | | 12 | | | 0 | | | 0 | | |
| Q14 | Ethos | 41 | 2.45 | 0.117 | 37 | 0.12 | 0.732 | 2 | 0.33 | 0.563 | 1 | 0.00 | 1 |
| | Logos | 28 | | | 40 | | | 1 | | | 1 | | |
| Q15 | Ethos | 34 | 0.01 | 0.904 | 33 | 1.57 | 0.21 | 1 | 0.33 | 0.563 | 1 | 0.00 | 1 |
| | Aesthetics | 35 | | | 44 | | | 2 | | | 1 | | |
| Q16 | Pathos | 36 | 0.13 | 0.717 | 26 | 8.12 | 0.004 | 2 | 0.33 | 0.563 | 1 | 0.00 | 1 |
| | Logos | 33 | | | 51 | | | 1 | | | 1 | | |
| Q17 | Pathos | 13 | 26.80 | <0.001 | 15 | 28.69 | <0.001 | 0 | 3.00 | 0.083 | 0 | 2.00 | 0.157 |
| | Aesthetics | 56 | | | 62 | | | 3 | | | 2 | | |
| Q18 | Logos | 25 | 5.23 | 0.022 | 34 | 1.05 | 0.305 | 1 | 0.33 | 0.563 | 1 | 0.00 | 1 |
| | Aesthetics | 44 | | | 43 | | | 2 | | | 1 | | |
| Q19 | Positive Reinforcement | 41 | 2.45 | 0.117 | 41 | 0.33 | 0.568 | 3 | 3.00 | 0.083 | 0 | 2.00 | 0.157 |
| | Positive Punishment | 28 | | | 36 | | | 0 | | | 2 | | |
| Q20 | Positive Reinforcement | 39 | 1.17 | 0.278 | 51 | 8.12 | 0.004 | 2 | 0.33 | 0.563 | 0 | 2.00 | 0.157 |
| | Negative Punishment | 30 | | | 26 | | | 1 | | | 2 | | |
| Q21 | Positive Reinforcement | 30 | 1.17 | 0.278 | 32 | 2.20 | 0.138 | 2 | 0.33 | 0.563 | 2 | 2.00 | 0.157 |
| | Negative Reinforcement | 39 | | | 45 | | | 1 | | | 0 | | |
| Q22 | Positive Punishment | 33 | 0.13 | 0.717 | 53 | 10.92 | 0.001 | 0 | 3.00 | 0.083 | 1 | 0.00 | 1 |
| | Negative Punishment | 36 | | | 24 | | | 3 | | | 1 | | |
| Q23 | Positive Punishment | 27 | 3.26 | 0.070 | 36 | 0.33 | 0.568 | 0 | 3.00 | 0.083 | 1 | 0.00 | 1 |
| | Negative Reinforcement | 42 | | | 41 | | | 3 | | | 1 | | |
| Q24 | Negative Punishment | 18 | 15.78 | 0.001 | 22 | 14.14 | 0.001 | 1 | 0.33 | 0.563 | 1 | 0.00 | 1 |
| | Negative Reinforcement | 51 | | | 55 | | | 2 | | | 1 | | |
| Q25 | Positive Reinforcement | 45 | 6.39 | 0.011 | 45 | 2.20 | 0.138 | 3 | 3.00 | 0.083 | 0 | 2.00 | 0.157 |
| | Positive Punishment | 24 | | | 32 | | | 0 | | | 2 | | |
| Q26 | Positive Reinforcement | 42 | 3.26 | 0.070 | 59 | 21.83 | <0.001 | 3 | 3.00 | 0.083 | 1 | 0.00 | 1 |
| | Negative Punishment | 27 | | | 18 | | | 0 | | | 1 | | |
| Q27 | Positive Reinforcement | 35 | 0.01 | 0.904 | 41 | 0.33 | 0.568 | 2 | 0.33 | 0.563 | 1 | 0.00 | 1 |
| | Negative Reinforcement | 34 | | | 36 | | | 1 | | | 1 | | |
| Q28 | Positive Punishment | 34 | 0.01 | 0.904 | 50 | 6.87 | 0.008 | 0 | 3.00 | 0.083 | 2 | 2.00 | 0.157 |
| | Negative Punishment | 35 | | | 27 | | | 3 | | | 0 | | |
| Q29 | Positive Punishment | 26 | 4.19 | 0.040 | 41 | 0.33 | 0.568 | 0 | 3.00 | 0.083 | 2 | 2.00 | 0.157 |
| | Negative Reinforcement | 43 | | | 36 | | | 3 | | | 0 | | |
| Q30 | Negative Punishment | 22 | 9.06 | 0.002 | 23 | 12.48 | 0.001 | 0 | 3.00 | 0.083 | 1 | 0.00 | 1 |
| | Negative Reinforcement | 47 | | | 54 | | | 3 | | | 1 | | |

**Table A3.** Analysis of multiple-choice questions based on awareness. (Green highlight indicates that the results are statistically significant at $p \leq 0.05$).

| | | Tremendous | | | High | | | Moderate | | | Little | | | Very Little | | |
|---|---|---|---|---|---|---|---|---|---|---|---|---|---|---|---|---|---|
| | | Score | Chi^2 Value | *p*-Value | Score | Chi^2 Value | *p*-Value | Score | Chi^2 Value | *p*-Value | Score | Chi^2 Value | *p*-Value | Score | Chi^2 Value | *p*-Value |
| Q7 | Ethos | 20 | 10.67 | 0.001 | 66 | 45.46 | <0.001 | 41 | 22.22 | <0.001 | 3 | 1.00 | 0.317 | 0 | - | - |
| | Pathos | 4 | | | 8 | | | 8 | | | 1 | | | 0 | | |
| Q8 | Ethos | 12 | 0.00 | 1 | 49 | 7.78 | 0.005 | 29 | 1.65 | 0.198 | 3 | 1.00 | 0.317 | 0 | - | - |
| | Logos | 12 | | | 25 | | | 20 | | | 1 | | | 0 | | |
| Q9 | Ethos | 12 | 0.00 | 1 | 31 | 1.95 | 0.163 | 20 | 1.65 | 0.198 | 3 | 1.00 | 0.317 | 0 | - | - |
| | Aesthetics | 12 | | | 43 | | | 29 | | | 1 | | | 0 | | |
| Q10 | Pathos | 9 | 1.50 | 0.220 | 40 | 0.49 | 0.485 | 25 | 0.02 | 0.886 | 2 | 0.00 | 1 | 0 | - | - |
| | Logos | 15 | | | 34 | | | 24 | | | 2 | | | 0 | | |
| Q11 | Pathos | 5 | 8.17 | 0.004 | 13 | 31.14 | <0.001 | 10 | 17.16 | 0.001 | 2 | 0.00 | 1 | 0 | - | - |
| | Aesthetics | 19 | | | 61 | | | 39 | | | 2 | | | 0 | | |
| Q12 | Logos | 7 | 4.17 | 0.041 | 20 | 15.62 | 0.001 | 15 | 7.37 | 0.006 | 1 | 1.00 | 0.317 | 0 | - | - |
| | Aesthetics | 17 | | | 54 | | | 34 | | | 3 | | | 0 | | |
| Q13 | Ethos | 20 | 10.67 | 0.001 | 66 | 45.46 | <0.001 | 41 | 22.22 | <0.001 | 2 | 0.00 | 1 | 0 | - | - |
| | Pathos | 4 | | | 8 | | | 8 | | | 2 | | | 0 | | |
| Q14 | Ethos | 12 | 0.00 | 1 | 43 | 1.95 | 0.163 | 23 | 0.18 | 0.668 | 3 | 1.00 | 0.317 | 0 | - | - |
| | Logos | 12 | | | 31 | | | 26 | | | 1 | | | 0 | | |
| Q15 | Ethos | 12 | 0.00 | 1 | 37 | 0.00 | 1 | 18 | 3.45 | 0.063 | 2 | 0.00 | 1 | 0 | - | - |
| | Aesthetics | 12 | | | 37 | | | 31 | | | 2 | | | 0 | | |
| Q16 | Pathos | 9 | 1.50 | 0.220 | 33 | 0.87 | 0.352 | 20 | 1.65 | 0.198 | 3 | 1.00 | 0.317 | 0 | - | - |
| | Logos | 15 | | | 41 | | | 29 | | | 1 | | | 0 | | |
| Q17 | Pathos | 5 | 8.17 | 0.004 | 14 | 28.60 | <0.001 | 7 | 25.00 | <0.001 | 2 | 0.00 | 1 | 0 | - | - |
| | Aesthetics | 19 | | | 60 | | | 42 | | | 2 | | | 0 | | |
| Q18 | Logos | 8 | 2.67 | 0.102 | 31 | 1.95 | 0.163 | 20 | 1.65 | 0.198 | 2 | 0.00 | 1 | 0 | - | - |
| | Aesthetics | 16 | | | 43 | | | 29 | | | 2 | | | 0 | | |
| Q19 | Positive Reinforcement | 12 | 0.00 | 1 | 44 | 2.65 | 0.103 | 28 | 1.00 | 0.317 | 1 | 1.00 | 0.317 | 0 | - | - |
| | Positive Punishment | 12 | | | 30 | | | 21 | | | 3 | | | 0 | | |

**Table A3.** *Cont.*

| | | Tremendous | | | High | | | Moderate | | | Little | | | Very Little | | |
|---|---|---|---|---|---|---|---|---|---|---|---|---|---|---|---|---|
| | | Score | Chi^2 Value | p-Value | Score | Chi^2 Value | p-Value | Score | Chi^2 Value | p-Value | Score | Chi^2 Value | p-Value | Score | Chi^2 Value | p-Value |
| Q20 | Positive Reinforcement | 15 | 1.50 | 0.220 | 43 | 1.95 | 0.163 | 33 | 5.90 | 0.015 | 1 | 1.00 | 0.317 | 0 | - | - |
| | Negative Punishment | 9 | | | 31 | | | 16 | | | 3 | | | 0 | | |
| Q21 | Positive Reinforcement | 6 | 6.00 | 0.014 | 37 | 0.00 | 1 | 22 | 0.51 | 0.475 | 1 | 1.00 | 0.317 | 0 | - | - |
| | Negative Reinforcement | 18 | | | 37 | | | 27 | | | 3 | | | 0 | | |
| Q22 | Positive Punishment | 12 | 0.00 | 1 | 40 | 0.49 | 0.485 | 33 | 5.90 | 0.015 | 2 | 0.00 | 1 | 0 | - | - |
| | Negative Punishment | 12 | | | 34 | | | 16 | | | 2 | | | 0 | | |
| Q23 | Positive Punishment | 8 | 2.67 | 0.102 | 33 | 0.87 | 0.352 | 21 | 1.00 | 0.317 | 2 | 0.00 | 1 | 0 | - | - |
| | Negative Reinforcement | 16 | | | 41 | | | 28 | | | 2 | | | 0 | | |
| Q24 | Negative Punishment | 3 | 13.50 | 0.001 | 23 | 10.60 | 0.001 | 15 | 7.37 | 0.006 | 1 | 1.00 | 0.317 | 0 | - | - |
| | Negative Reinforcement | 21 | | | 51 | | | 34 | | | 3 | | | 0 | | |
| Q25 | Positive Reinforcement | 15 | 1.50 | 0.220 | 49 | 7.78 | 0.005 | 27 | 0.51 | 0.475 | 2 | 0.00 | 1 | 0 | - | - |
| | Positive Punishment | 9 | | | 25 | | | 22 | | | 2 | | | 0 | | |
| Q26 | Positive Reinforcement | 15 | 1.50 | 0.220 | 51 | 10.60 | 0.001 | 36 | 10.80 | 0.001 | 3 | 1.00 | 0.317 | 0 | - | - |
| | Negative Punishment | 9 | | | 23 | | | 13 | | | 1 | | | 0 | | |
| Q27 | Positive Reinforcement | 10 | 0.67 | 0.414 | 40 | 0.49 | 0.485 | 28 | 1.00 | 0.317 | 1 | 1.00 | 0.317 | 0 | - | - |
| | Negative Reinforcement | 14 | | | 34 | | | 21 | | | 3 | | | 0 | | |
| Q28 | Positive Punishment | 13 | 0.17 | 0.683 | 39 | 0.22 | 0.641 | 31 | 3.45 | 0.063 | 3 | 1.00 | 0.317 | 0 | - | - |
| | Negative Punishment | 11 | | | 35 | | | 18 | | | 1 | | | 0 | | |
| Q29 | Positive Punishment | 9 | 1.50 | 0.220 | 33 | 0.87 | 0.352 | 25 | 0.02 | 0.886 | 2 | 0.00 | 1 | 0 | - | - |
| | Negative Reinforcement | 15 | | | 41 | | | 24 | | | 2 | | | 0 | | |
| Q30 | Negative Punishment | 3 | 13.50 | 0.001 | 23 | 10.60 | 0.001 | 18 | 3.45 | 0.063 | 2 | 0.00 | 1 | 0 | - | - |
| | Negative Reinforcement | 21 | | | 51 | | | 31 | | | 2 | | | 0 | | |

**Table A4.** Analysis of multiple-choice questions based on recycling efforts. (Green highlight indicates that the results are statistically significant at $p \leq 0.05$).

| | | Tremendous | | | High | | | Moderate | | | Little | | | Very Little | | |
|---|---|---|---|---|---|---|---|---|---|---|---|---|---|---|---|---|
| | | Score | Chi^2 Value | p-Value | Score | Chi^2 Value | p-Value | Score | Chi^2 Value | p-Value | Score | Chi^2 Value | p-Value | Score | Chi^2 Value | p-Value |
| Q7 | Ethos | 5 | 1.29 | 0.256 | 62 | 39.56 | <0.001 | 50 | 34.57 | <0.001 | 12 | 7.14 | 0.007 | 1 | 0.33 | 0.563 |
| | Pathos | 2 | | | 9 | | | 6 | | | 2 | | | 2 | | |
| Q8 | Ethos | 4 | 0.14 | 0.705 | 45 | 5.09 | 0.024 | 35 | 3.50 | 0.061 | 7 | 0.00 | 1 | 2 | 0.33 | 0.563 |
| | Logos | 3 | | | 26 | | | 21 | | | 7 | | | 1 | | |
| Q9 | Ethos | 4 | 0.14 | 0.705 | 28 | 3.17 | 0.075 | 29 | 0.07 | 0.789 | 3 | 4.57 | 0.032 | 2 | 0.33 | 0.563 |
| | Aesthetics | 3 | | | 43 | | | 27 | | | 11 | | | 1 | | |
| Q10 | Pathos | 3 | 0.14 | 0.705 | 35 | 0.01 | 0.905 | 31 | 0.64 | 0.422 | 5 | 1.14 | 0.285 | 2 | 0.33 | 0.563 |
| | Logos | 4 | | | 36 | | | 25 | | | 9 | | | 1 | | |
| Q11 | Pathos | 2 | 1.29 | 0.256 | 13 | 28.52 | <0.001 | 11 | 20.64 | <0.001 | 2 | 7.14 | 0.007 | 2 | 0.33 | 0.563 |
| | Aesthetics | 5 | | | 58 | | | 45 | | | 12 | | | 1 | | |
| Q12 | Logos | 3 | 0.14 | 0.705 | 18 | 7.25 | 0.001 | 17 | 8.64 | 0.003 | 4 | 2.57 | 0.108 | 1 | 0.33 | 0.563 |
| | Aesthetics | 4 | | | 53 | | | 39 | | | 10 | | | 2 | | |
| Q13 | Ethos | 5 | 1.29 | 0.256 | 63 | 42.61 | <0.001 | 49 | 31.50 | <0.001 | 11 | 4.57 | 0.032 | 1 | 0.33 | 0.563 |
| | Pathos | 2 | | | 8 | | | 7 | | | 3 | | | 2 | | |
| Q14 | Ethos | 3 | 0.14 | 0.705 | 35 | 0.01 | 0.905 | 34 | 2.57 | 0.108 | 7 | 0.00 | 1 | 2 | 0.33 | 0.563 |
| | Logos | 4 | | | 36 | | | 22 | | | 7 | | | 1 | | |
| Q15 | Ethos | 4 | 0.14 | 0.705 | 32 | 0.69 | 0.406 | 29 | 0.07 | 0.789 | 2 | 7.14 | 0.007 | 2 | 0.33 | 0.563 |
| | Aesthetics | 3 | | | 39 | | | 27 | | | 12 | | | 1 | | |
| Q16 | Pathos | 3 | 0.14 | 0.705 | 25 | 6.21 | 0.012 | 30 | 0.29 | 0.592 | 5 | 1.14 | 0.285 | 2 | 0.33 | 0.563 |
| | Logos | 4 | | | 46 | | | 26 | | | 9 | | | 1 | | |
| Q17 | Pathos | 2 | 1.29 | 0.256 | 12 | 31.11 | <0.001 | 10 | 23.14 | <0.001 | 2 | 7.14 | 0.007 | 2 | 0.33 | 0.563 |
| | Aesthetics | 5 | | | 59 | | | 46 | | | 12 | | | 1 | | |
| Q18 | Logos | 4 | 0.14 | 0.705 | 26 | 5.09 | 0.024 | 23 | 1.79 | 0.181 | 7 | 0.00 | 1 | 1 | 0.33 | 0.563 |
| | Aesthetics | 3 | | | 45 | | | 33 | | | 7 | | | 2 | | |

**Table A4.** *Cont.*

| | | Tremendous | | | High | | | Moderate | | | Little | | | Very Little | | |
|---|---|---|---|---|---|---|---|---|---|---|---|---|---|---|---|---|
| | | Score | Chi^2 Value | p-Value | Score | Chi^2 Value | p-Value | Score | Chi^2 Value | p-Value | Score | Chi^2 Value | p-Value | Score | Chi^2 Value | p-Value |
| Q19 | Positive Reinforcement | 6 | 3.57 | 0.587 | 40 | 1.14 | 0.285 | 30 | 0.29 | 0.592 | 7 | 0.00 | 1 | 2 | 0.33 | 0.563 |
| | Positive Punishment | 1 | | | 31 | | | 26 | | | 7 | | | 1 | | |
| Q20 | Positive Reinforcement | 4 | 0.14 | 0.705 | 42 | 2.38 | 0.122 | 33 | 1.79 | 0.181 | 11 | 4.57 | **0.032** | 2 | 0.33 | 0.563 |
| | Negative Punishment | 3 | | | 29 | | | 23 | | | 3 | | | 1 | | |
| Q21 | Positive Reinforcement | 4 | 0.14 | 0.705 | 32 | 0.69 | 0.406 | 21 | 3.50 | 0.061 | 7 | 0.00 | 1 | 2 | 0.33 | 0.563 |
| | Negative Reinforcement | 3 | | | 39 | | | 35 | | | 7 | | | 1 | | |
| Q22 | Positive Punishment | 3 | 0.14 | 0.705 | 39 | 0.69 | 0.406 | 37 | 5.79 | **0.016** | 7 | 0.00 | 1 | 1 | 0.33 | 0.563 |
| | Negative Punishment | 4 | | | 32 | | | 19 | | | 7 | | | 2 | | |
| Q23 | Positive Punishment | 2 | 1.29 | 0.256 | 29 | 2.38 | 0.122 | 27 | 0.07 | 0.789 | 6 | 0.29 | 0.592 | 0 | 3.00 | 0.083 |
| | Negative Reinforcement | 5 | | | 42 | | | 29 | | | 8 | | | 3 | | |
| Q24 | Negative Punishment | 0 | 7.00 | **0.008** | 15 | 23.68 | **<0.001** | 19 | 5.79 | **0.016** | 8 | 0.29 | 0.592 | 0 | 3.00 | 0.083 |
| | Negative Reinforcement | 7 | | | 56 | | | 37 | | | 6 | | | 3 | | |
| Q25 | Positive Reinforcement | 6 | 3.57 | 0.587 | 44 | 4.07 | **0.043** | 33 | 1.79 | 0.181 | 7 | 0.00 | 1 | 3 | 3.00 | 0.083 |
| | Positive Punishment | 1 | | | 27 | | | 23 | | | 7 | | | 0 | | |
| Q26 | Positive Reinforcement | 4 | 0.14 | 0.705 | 50 | 11.85 | **0.001** | 38 | 7.14 | **0.007** | 10 | 2.57 | 0.108 | 3 | 3.00 | 0.083 |
| | Negative Punishment | 3 | | | 21 | | | 18 | | | 4 | | | 0 | | |
| Q27 | Positive Reinforcement | 4 | 0.14 | 0.705 | 38 | 0.35 | 0.552 | 26 | 0.29 | 0.592 | 9 | 1.14 | 0.285 | 2 | 0.33 | 0.563 |
| | Negative Reinforcement | 3 | | | 33 | | | 30 | | | 5 | | | 1 | | |
| Q28 | Positive Punishment | 3 | 0.14 | 0.705 | 39 | 0.69 | 0.406 | 36 | 4.57 | **0.032** | 6 | 0.29 | 0.592 | 2 | 0.33 | 0.563 |
| | Negative Punishment | 4 | | | 32 | | | 20 | | | 8 | | | 1 | | |
| Q29 | Positive Punishment | 1 | 3.57 | 0.587 | 30 | 1.70 | 0.191 | 30 | 0.29 | 0.592 | 8 | 0.29 | 0.592 | 0 | 3.00 | 0.083 |
| | Negative Reinforcement | 6 | | | 41 | | | 26 | | | 6 | | | 3 | | |
| Q30 | Negative Punishment | 1 | 3.57 | 0.587 | 17 | 19.28 | **0.001** | 19 | 5.79 | **0.016** | 9 | 1.14 | 0.285 | 0 | 3.00 | 0.083 |
| | Negative Reinforcement | 6 | | | 54 | | | 37 | | | 5 | | | 3 | | |

**Table A5.** Results of Likert scale questions based on age.

| | 18–30 | 31–45 | 46+ | Prefer Not to Answer |
|---|---|---|---|---|
| Aesthetics | 4.39 | 4.30 | 4.14 | 3.84 |
| Ethos | 4.40 | 4.25 | 3.96 | 3.56 |
| Logos | 3.63 | 3.432 | 3.26 | 3.32 |
| Pathos | 4.05 | 4.05 | 3.87 | 3.32 |

**Table A6.** Analysis of Likert scale questions based on age. (Green highlight indicates that the results are statistically significant at $p \le 0.05$).

| Comparison of Persuasion Techniques | 18–30 | | 31–45 | | 46+ | | Prefer Not to Answer | |
|---|---|---|---|---|---|---|---|---|
| | t-Value | p-Value | t-Value | p-Value | t-Value | p-Value | t-Value | p-Value |
| Ethos with Pathos | 2.47 | **0.019** | 1.69 | 0.063 | 0.6957 | 0.2532 | 1.54 | 0.080 |
| Ethos with Logos | 1.80 | 0.054 | 2.42 | **0.020** | 2.41 | **0.021** | 0.55 | 0.297 |
| Aesthetics with Ethos | 0.10 | 0.460 | 0.82 | 0.217 | 2.57 | **0.016** | 1.19 | 0.133 |
| Aesthetics with Pathos | 2.57 | **0.016** | 2.04 | **0.037** | 2.14 | **0.032** | 1.99 | **0.040** |
| Aesthetics with Logos | 1.80 | 0.054 | 2.57 | **0.016** | 3.07 | **0.007** | 1.07 | 0.155 |
| Logos with Pathos | 0.96 | 0.181 | 0.75 | 0.059 | 1.95 | **0.042** | 0.00 | 0.500 |

**Table A7.** Results of Likert scale questions based on gender.

|  | **Male** | **Female** | **Non-Binary** | **Prefer Not to Mention** |
|---|---|---|---|---|
| Aesthetics | 4.26 | 4.31 | 4.30 | 3.80 |
| Ethos | 4.13 | 4.29 | 4.30 | 3.66 |
| Logos | 3.46 | 3.47 | 2.90 | 3.13 |
| Pathos | 3.93 | 4.04 | 3.90 | 3.46 |

**Table A8.** Analysis of Likert scale questions based on gender. (Green highlight indicates that the results are statistically significant at $p \leq 0.05$).

| **Comparison of Persuasion Techniques** | **Male** | | **Female** | | **Non-Binary** | | **Prefer Not to Mention** | |
|---|---|---|---|---|---|---|---|---|
|  | *t*-Value | *p*-Value | *t*-Value | *p*-Value | *t*-Value | *p*-Value | *t*-Value | *p*-Value |
| Ethos with Pathos | 1.59 | 0.075 | 1.94 | 0.044 | 2.52 | 0.017 | 1.50 | 0.086 |
| Ethos with Logos | 2.50 | 0.018 | 1.91 | 0.045 | 2.45 | 0.019 | 0.96 | 0.181 |
| Aesthetics with Ethos | 2.15 | 0.031 | 0.30 | 0.385 | 0.00 | 0.500 | 1.00 | 0.173 |
| Aesthetics with Pathos | 2.50 | 0.018 | 2.24 | 0.027 | 2.52 | 0.017 | 1.76 | 0.057 |
| Aesthetics with Logos | 2.95 | 0.009 | 1.97 | 0.041 | 2.45 | 0.019 | 1.17 | 0.137 |
| Logos with Pathos | 1.60 | 0.073 | 1.3060 | 0.1139 | 1.76 | 0.057 | 0.58 | 0.287 |

**Table A9.** Results of Likert scale questions based on awareness.

|  | **Tremendous** | **High** | **Moderate** | **Little** |
|---|---|---|---|---|
| Aesthetics | 4.44 | 4.28 | 4.15 | 4.75 |
| Ethos | 4.51 | 4.22 | 4.02 | 4.50 |
| Logos | 3.27 | 3.50 | 3.46 | 3.55 |
| Pathos | 4.08 | 4.05 | 3.82 | 4.00 |

**Table A10.** Analysis of Likert scale questions based on awareness. (Green highlight indicates that the results are statistically significant at $p \leq 0.05$).

| **Comparison of Persuasion Techniques** | **Tremendous** | | **High** | | **Moderate** | | **Little** | |
|---|---|---|---|---|---|---|---|---|
|  | *t*-Value | *p*-Value | *t*-Value | *p*-Value | *t*-Value | *p*-Value | *t*-Value | *p*-Value |
| Ethos with Pathos | 3.18 | 0.006 | 1.64 | 0.069 | 1.19 | 0.132 | 1.58 | 0.076 |
| Ethos with Logos | 3.08 | 0.007 | 2.09 | 0.034 | 1.51 | 0.084 | 3.16 | 0.006 |
| Aesthetics with Ethos | 1.13 | 0.144 | 1.24 | 0.123 | 1.77 | 0.057 | 1.82 | 0.052 |
| Aesthetics with Pathos | 2.68 | 0.013 | 2.27 | 0.026 | 2.10 | 0.034 | 2.17 | 0.030 |
| Aesthetics with Logos | 2.90 | 0.009 | 2.28 | 0.025 | 1.90 | 0.046 | 3.63 | 0.003 |
| Logos with Pathos | 1.92 | 0.045 | 1.55 | 0.079 | 0.92 | 0.191 | 1.03 | 0.166 |

**Table A11.** Results of Likert scale questions based on recycling efforts.

|  | **Tremendous** | **High** | **Moderate** | **Little** | **Very Little** |
|---|---|---|---|---|---|
| Aesthetics | 4.00 | 4.44 | 4.17 | 3.97 | 4.73 |
| Ethos | 3.68 | 4.42 | 4.02 | 3.97 | 5.00 |
| Logos | 2.91 | 3.46 | 3.54 | 3.42 | 3.00 |
| Pathos | 3.74 | 4.20 | 3.82 | 3.42 | 4.86 |

**Table A12.** Analysis of Likert scale questions based on recycling efforts. (Green highlight indicates that the results are statistically significant at $p \leq 0.05$).

| Comparison of Persuasion Techniques | Tremendous | | High | | Moderate | | Little | | Very Little | |
|---|---|---|---|---|---|---|---|---|---|---|
| | *t*-Value | *p*-Value | *t*-Value | *p*-Value | *t*-Value | *p*-Value | *t*-Value | *p*-Value | *t*-Value | *p*-Value |
| Ethos with Pathos | 0.48 | 0.320 | 1.77 | 0.056 | 1.37 | 0.103 | 3.55 | 0.003 | 1.00 | 0.173 |
| Ethos with Logos | 1.53 | 0.081 | 2.40 | 0.021 | 1.50 | 0.085 | 1.65 | 0.068 | 7.17 | 0.001 |
| Aesthetics with Ethos | 2.55 | 0.016 | 0.23 | 0.411 | 2.77 | 0.012 | 0.00 | 0.500 | 1.37 | 0.103 |
| Aesthetics with Pathos | 2.44 | 0.020 | 1.88 | 0.048 | 2.40 | 0.021 | 2.48 | 0.018 | 0.56 | 0.293 |
| Aesthetics with Logos | 2.17 | 0.030 | 2.44 | 0.020 | 1.97 | 0.042 | 1.49 | 0.087 | 5.09 | 0.001 |
| Logos with Pathos | 1.66 | 0.067 | 1.80 | 0.054 | 0.81 | 0.219 | 0.00 | 0.500 | 6.03 | 0.001 |

**Appendix B**

Survey questionnaires

Q1 Definitions: Recycling process—You place the cardboard box in the dedicated recycle bin or return it to the dedicated recycling yard, which is then recycled to make a new cardboard box. Reusing process—You place the cardboard box in the dedicated reuse bin or return it to the dedicated reuse yard, where it is reused for shipping goods, and then the cardboard box is cleaned and prepared for another use.

- I understood the difference between these two processes.

Q2 Please enter your email id—_______________________

Q3 What gender do you identify as?

- Male
- Female
- Non-binary
- Prefer not to answer

Q4 What is your age?

- 0–17 years old
- 18–30 years old
- 31–45 years old
- 46+
- Prefer not to answer

Q5 What are your current recycling efforts?

- Very Little
- Little
- Moderate
- High
- Tremendous

Q6 How much awareness do you have of the environment and climate change?

- Very Little
- Little
- Moderate
- High
- Tremendous

Q7 Which one is more likely to influence you for assigning the cardboard box to the reuse bin rather than the recycling bin–

- A charitable organization committed to preventing environmental degradation gets a suitable donation for each box I assign to the reusing process.
- A charitable organization committed to helping Florida panthers from going extinct gets a suitable donation for each box I assign to the reusing process.

Q8 Which one is more likely to influence you for assigning the cardboard box to the reuse bin rather than the recycling bin–

- A charitable organization committed to preventing environmental degradation gets a suitable donation for each box I assign to the reusing process.
- I get a suitable cash reward for each box I assign to the reusing process.

Q9 Which one is more likely to influence you for assigning the cardboard box to the reuse bin rather than the recycling bin–

- A charitable organization committed to preventing environmental degradation gets a suitable donation for each box I assign to the reusing process.
- A charitable organization committed to keeping my city clean gets a suitable donation for each box I assign to the reusing process.

Q10 Which one is more likely to influence you for assigning the cardboard box to the reuse bin rather than the recycling bin–

- A charitable organization committed to helping Florida panthers from going extinct gets a suitable donation for each box I assign to the reusing process.
- I get a suitable cash reward for each box I assign to the reusing process.

Q11 Which one is more likely to influence you for assigning the cardboard box to the reuse bin rather than the recycling bin–

- A charitable organization committed to helping Florida panthers from going extinct gets a suitable donation for each box I assign to the reusing process.
- A charitable organization committed to keeping my city clean gets a suitable donation for each box I assign to the reusing process.

Q12 Which one is more likely to influence you for assigning the cardboard box to the reuse bin rather than the recycling bin–

- I get a suitable cash reward for each box I assign to the reusing process.
- A charitable organization committed to keeping my city clean gets a suitable donation for each box I assign to the reusing process.

Q13 Which one is more likely to influence you for returning the cardboard box to the reuse yard rather than the recycling yard –

- A charitable organization committed to preventing environmental degradation gets a suitable donation for each box I return to the reuse yard.
- A charitable organization committed to helping Florida panthers from going extinct gets a suitable donation for each box I return to the reuse yard.

Q14 Which one is more likely to influence you for returning the cardboard box to the reuse yard rather than the recycling yard –

- A charitable organization committed to preventing environmental degradation gets a suitable donation for each box I return to the reuse yard.
- I get a suitable cash reward for each box I return to the reuse yard.

Q15 Which one is more likely to influence you for returning the cardboard box to the reuse yard rather than the recycling yard –

- A charitable organization committed to preventing environmental degradation gets a suitable donation for each box I return to the reuse yard.
- A charitable organization committed to keeping my city clean gets a suitable donation for each box I return to the reuse yard.

Q16 Which one is more likely to influence you for returning the cardboard box to the reuse yard rather than the recycling yard –

- A charitable organization committed to helping Florida panthers from going extinct gets a suitable donation for each box I return to the reuse yard.
- I get a suitable cash reward for each box I return to the reuse yard.

Q17 Which one is more likely to influence you for returning the cardboard box to the reuse yard rather than the recycling yard –

- A charitable organization committed to helping Florida panthers from going extinct gets a suitable donation for each box I return to the reuse yard.
- A charitable organization committed to keeping my city clean gets a suitable donation for each box I return to the reuse yard.

Q18 Which one is more likely to influence you for returning the cardboard box to the reuse yard rather than the recycling yard –

- I get a suitable cash reward for each box I return to the reuse yard.
- A charitable organization committed to keeping my city clean gets a suitable donation for each box I return to the reuse yard.

Q19 Which one is more likely to influence you for assigning the cardboard box to the reuse bin rather than the recycling bin–

- I get a suitable cash reward for each box I assign to the reuse process.
- I get penalized with a suitable cash penalty for not assigning the boxes to the reuse process.

Q20 Which one is more likely to influence you for assigning the cardboard box to the reuse bin rather than the recycling bin–

- I get a suitable cash reward for each box I assign to the reuse process.
- My product discount is taken away from me which was offered to me for every cardboard box I assign to the reuse process.

Q21 Which one is more likely to influence you for assigning the cardboard box to the reuse bin rather than the recycling bin–

- I get a suitable cash reward for each box I assign to the reuse process.
- My shipping charges are waived after I assign a suitable number of boxes to the reuse process.

Q22 Which one is more likely to influence you for assigning the cardboard box to the reuse bin rather than the recycling bin–

- I get penalized with a suitable cash penalty for not assigning the boxes to the reuse process.
- My product discount is taken away from me which was offered to me for every cardboard box I assign to the reuse process.

Q23 Which one is more likely to influence you for assigning the cardboard box to the reuse bin rather than the recycling bin–

- I get penalized with a suitable cash penalty for not assigning the boxes for the reusing process.
- My shipping charges are waived after I assign a suitable number of boxes to the reuse process.

Q24 Which one is more likely to influence you for assigning the cardboard box to the reuse bin rather than the recycling bin–

- My product discount is taken away from me which was offered to me for every cardboard box I assign to the reuse process.
- My shipping charges are waived after I assign a suitable number of boxes to the reuse process.

Q25 Which one is more likely to influence you for returning the cardboard box to the reuse yard rather than the recycling yard –

- I get a suitable cash reward for each box I return to the reuse yard.
- I get penalized with a suitable cash penalty for not returning the boxes to the reuse yard.

Q26 Which one is more likely to influence you for returning the cardboard box to the reuse yard rather than the recycling yard –

- I get a suitable cash reward for each box I return to the reuse yard.

- My product discount is taken away from me which was offered to me for every cardboard box I return to the reuse yard.

Q27 Which one is more likely to influence you for returning the cardboard box to the reuse yard rather than the recycling yard –

- I get a suitable cash reward for each box I return to the reuse yard.
- My shipping charges are waived after I return a suitable number of boxes to the reuse yard.

Q28 Which one is more likely to influence you for returning the cardboard box to the reuse yard rather than the recycling yard –

- I get penalized with a suitable cash penalty for not returning the boxes to the reuse yard.
- My product discount is taken away from me which was offered to me for every cardboard box I return to the reuse yard.

Q29 Which one is more likely to influence you for returning the cardboard box to the reuse yard rather than the recycling yard –

- I get penalized with a suitable cash penalty for not returning the boxes to the reuse yard.
- My shipping charges are waived after I return a suitable number of boxes to the reuse yard.

Q30 Which one is more likely to influence you for returning the cardboard box to the reuse yard rather than the recycling yard –

- My product discount is taken away from me which was offered to me for every cardboard box I return to the reuse yard.
- My shipping charges are waived after I return a suitable number of boxes to the reuse yard.

Q31 I am likely to assign a cardboard box to the reuse process rather than assigning it to the recycling process if—(Strongly disagree, Somewhat disagree, Neither agree nor disagree, Somewhat agree, and Strongly agree) (NO QUESTION)

Q32 A charitable organization committed to preventing environmental degradation gets a suitable donation for each box I assign to the reuse process.

Q33 A charitable organization committed to helping Florida panthers from going extinct gets a suitable donation for each box I assign to the reuse process.

Q34 I get a suitable cash reward for each box I assign to the reuse process.

Q35 A charitable organization committed to keeping my city clean gets a suitable donation for each box I assign to the reuse process.

Q36 A charitable organization trying to reduce global warming gets a suitable donation for each box I assign to the reuse process.

Q37 I am likely to assign a cardboard box to the reuse process rather than assigning it to the recycling process if—(Strongly disagree, Somewhat disagree, Neither agree nor disagree, Somewhat agree, and Strongly agree) (NO QUESTION)

Q38 A charitable organization trying to repair the ozone layer gets a suitable donation for each box I assign to the reuse process.

Q39 I save money off my shipping charges for each box I assign to the reuse process.

Q40 A charitable organization committed to preventing the addition of trash into landfills gets a suitable donation for each box I assign to the reuse process.

Q41 A charitable organization committed to reducing pollution gets a suitable donation for each box I assign to the reuse process.

Q42 A charitable organization committed to helping polar bears from going extinct gets a suitable donation for each box I assign to the reuse process.

Q43 I am likely to assign a cardboard box to the reuse process rather than assigning it to the recycling process if—(Strongly disagree, Somewhat disagree, Neither agree nor disagree, Somewhat agree, and Strongly agree) (NO QUESTION)

Q44 I get a suitable discount on my favorite shopping brands for each box I assign to the reuse process.

Q45 A charitable organization committed to cleaning the trash in my city gets a suitable donation for each box I assign to the reuse process.

Q46 A charitable organization trying to decrease the depletion of fossil fuel gets a suitable donation for each box I assign to the reuse process.

Q47 A charitable organization committed to helping endangered species gets a suitable donation for each box I assign to the reuse process.

Q48 I get public recognition after I assign a suitable number of boxes to the reuse process.

Q49 I am likely to assign a cardboard box to the reuse process rather than assigning it to the recycling process if—(Strongly disagree, Somewhat disagree, Neither agree nor disagree, Somewhat agree, and Strongly agree) (NO QUESTION)

Q50 A charitable organization committed to keeping our environment clean gets a suitable donation for each box I assign to the reuse process.

Q51 A charitable organization committed to reducing climate change gets a suitable donation for each box I assign to the reuse process.

Q52 A charitable organization committed to preserving the environment for future generations gets a suitable donation for each box I assign to the reuse process.

Q53 I get a gift card for my favorite fast-food brand for each box I assign to the reuse process.

Q54 A charitable organization committed to decreasing dirty landfills gets a suitable donation for each box I assign to the reuse process.

Q55 I prefer driving sustainable electric cars over gasoline-powered cars.

Q56 I prefer environment-friendly fabric bags over cheap plastic bags in grocery stores.

Q57 I routinely donate food/money to the less fortunate.

Q58 I work hard to receive praise from my boss.

Q59 I avoid losing important documents by organizing them in the first place.

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
