# Peer review of "Demographic Considerations in Incenting Reuse of Corrugated Cardboard Boxes"

_sustainability, doi:10.3390/su151511600_

Round 1

Reviewer 1 Report

This manuscript deals with a study about demographic considerations in incenting reuse of corrugated cardboard boxes. 

The manuscript is interesting and relevant for the Sustainibility journal. It is well written and well structured. I have inly a few very minor comments:

·        Abstract line 12: The word ”also” is unclear. You have not mentioned anything about this paper previously.

·        Line 46: The references 5-9 should be mentioned in clear text.

·        Line 88-89: It would be easier to read the manuscript if there is a short explanation of ”entropy calculations”

·        Line 222. It should be Table 12, not Table 92.

Author Response

Comments from reviewers and feedback from authors are as below -

This manuscript deals with a study about demographic considerations in incenting reuse of corrugated cardboard boxes. 

The manuscript is interesting and relevant for the Sustainibility journal. It is well written and well structured. I have inly a few very minor comments:

  • Abstract line 12: The word ”also” is unclear. You have not mentioned anything about this paper previously.

Thank you very much for bringing this to our attention, we have removed the word “also”. (line 12)

  • Line 46: The references 5-9 should be mentioned in clear text.

Thank you for your comment, Lines 43-46 are changed to “Although recycling would help in reducing GHG emissions, the motivation for recycling is lacking in the general population it has been observed by Abila [11]; Gilli et al. [12]; Kattoua et al. [13]; Seacat and Boileau [14]; and Li et al. [15].”

  • Line 88-89: It would be easier to read the manuscript if there is a short explanation of ”entropy calculations”

Thank you for your comment, the definition of entropy is added in lines 118-119 by adding the sentence “Entropy is a measure of randomness [16], random data gives high entropy and vice-versa.”

  • Line 222. It should be Table 12, not Table 92.

Thank you for your comment, this correction is made in line 253 by changing 92 to 12.

Reviewer 2 Report

Please see the comments in the attached document.

Author Response

Comments from reviewers and feedback from authors are as below –

Authors should pay attention to the revisions, addressing each and every comment.

  1. In introduction, in my opinion, the information that is handled there is very poor, the references are scarce, it requires a more in-depth investigation.

Thank you for your feedback! This comment is addressed by adding additional context and references in the introduction section in lines 39 to 47.

  1. It is necessary to explain in detail the materials and the methodology, since as it is present, it is not clear at all

These comments are addressed by rewriting the section. Lines 105-155

3. The relevance of this work is not demonstrated, it does not provide scientific knowledge.

The relevance/need of this work is demonstrated in the introduction section in lines 39 to 47.

  1. Conducting surveys is not relevant for an international manuscript.

Thank you for your comment, the survey conducted in this research has received participation at the international level (participants from 7 countries and 4 continents)

  1. No care has been taken in the numbering of the tables, from table 11 we skipped to 92.

Thank you for bringing this to our attention, this comment is addressed in line 253 by correcting the word “Table 92” to “Table 12”

  1. Figure 1, (a,b,c,d) is not well defined, please improve the presentation, the nomenclature of the x and y axes is not visible.

We have addressed this comment by adding Axis titles to the graphs. (lines 214-217)

  1. The discussion is very poor.

Dear Reviewer, in order to address this comment, we have change the Discussion section mainly in lines 304-340

  1. Conclusions in this version plays a role of summing up discussion and thus is too long. I suggest revising the text, adding the most important discussion into the Results and Discussion section, while rewriting the Conclusions with outlooks and possibilities scaling up the findings of this paper.

We have addressed this comment by shifting the text from the conclusion section to the discussion section and rewriting both sections. (lines 304-370)

  1. The appendices are put without further explanation; therefore, it does not make much sense to put so much data. It would be better to summarize them and make more concrete and clear graphs.

The reference for Appendix A is given in lines 219 to 221. Because the tables are big and contain a lot of data, they are in Appendix A. Whereas a summary of that data is given in Table 9 from lines 227-230.

10.The Appendix, from table A1 to table A 12, the number of significant digits should be double-checked and corrected throughout the text, the techniques and procedures used may have only one significant digit. Thus, the signal values should be changed in accordance with the error values. In the present version of the manuscript, all the values have too many significant digits.

Thank you for bringing this to our attention, this comment is addressed by changing the Chi^2 value/t-value to two decimal points (significant digits), and the p-values are changed to 3 decimal points (significant digits).

11.Appendix B, there is no point in putting it in an international article.

Thank you for the comment, we wanted to disclose the questionnaire with the paper so that the research study can be replicated or use the questionnaire we used in this paper.

12.References are extremely poor, also, please pay special attention to formatting of references in the Instructions to Authors. It is recommended to put the DOI in all referenced journals, it should be placed before the Appendixes.

We have added additional references to address these comments accordingly. Also, the reference section is moved above the Appendix section.

Reviewer 3 Report

Dear authors

I reviewed the manuscript entitled “Demographic considerations in incenting reuse of corrugated cardboard boxes”. This study is of great and important topic. In my opinion, it is greatly fall within the scope of journal. It should also be mentioned that the manuscript has addressed an issue that is my favorite subject area. Reading and evaluating different parts of the paper shows that the authors have made a lot of effort to carry out this research endeavor. Their efforts have resulted in important and ground-breaking conclusions that can certainly be used by different end-users including policy-makers, decision-makers, development authorities, practitioners, and so on. The authors have also used state of the art methodologies for analyzing their data. This makes their results and conclusions more reliable and rigorous. Therefore, I recommend this paper for publication. However, there are some points that should be addressed by the respected authors before consideration of the manuscript for publication in Sustainability. My main comments are as follows:

1.       In the abstract, the respected authors have mentioned that “The authors have previously research identifying the need to motivate the general population for sustainability efforts. The paper also proposes to use positive reinforcement ethos as a psychological incentive to motivate the general population”. This part needs to be reworded.

2.       One of the most important contributions of present study to the body of knowledge should be mentioned in the end of introduction.

3.       In the end of introduction, the main research questions (sub-objectives) should be mentioned.

4.       Please highlight the most important originalities of the research in the end introduction.

5.       This study has been done in The United States. However, I recommend the respected authors to highlight the global value of this research in the end of introduction section. Can the results of this study be applied by the users from other parts of the world?

6.       This paper has not the background. I am sure that there are some relative background studies around the world. Please review the literature in a separate section entitled “Background of the study”.

7.       Methods section is very short and naïve. Please elaborate this section using adding some sub-sections explaining:

-          Research design

-          Population and sampling approach

-          Research tool reliability and validity

-          Measures

-          Data analysis

8.       Results section has been written and articulated very well. There is no need for further revisions.

9.       In discussion section please try to put your results in an international scope and then provide the readers with some useful global level recommendations. Also, in discussion section the respected authors should try compare their results with the results of other researchers in the USA and other parts of the world.

10.   In conclusion section, I recommend the respected authors to mention the main take-home message of the research in a short paragraph.

11.   Please highlight the main limitation of your study and try to draw some future pathways for the future researchers.

12.   In conclusion section try to highlight the main contribution of your paper to the theory and practice.

In general, I believe that this manuscript can be accepted for publication in Sustainability after MAJOR revisions.

Author Response

Comments from reviewers and feedback from authors are as below –

Dear authors

I reviewed the manuscript entitled “Demographic considerations in incenting reuse of corrugated cardboard boxes”. This study is of great and important topic. In my opinion, it is greatly fall within the scope of journal. It should also be mentioned that the manuscript has addressed an issue that is my favorite subject area. Reading and evaluating different parts of the paper shows that the authors have made a lot of effort to carry out this research endeavor. Their efforts have resulted in important and ground-breaking conclusions that can certainly be used by different end-users including policy-makers, decision-makers, development authorities, practitioners, and so on. The authors have also used state of the art methodologies for analyzing their data. This makes their results and conclusions more reliable and rigorous. Therefore, I recommend this paper for publication. However, there are some points that should be addressed by the respected authors before consideration of the manuscript for publication in Sustainability. My main comments are as follows:

  1. In the abstract, the respected authors have mentioned that “The authors have previously research identifying the need to motivate the general population for sustainability efforts. The paper also proposes to use positive reinforcement ethos as a psychological incentive to motivate the general population”. This part needs to be reworded.

Dear reviewer, we have addressed this comment by changing the structure of the sentence and updating it in lines 10-11.

  1. One of the most important contributions of present study to the body of knowledge should be mentioned in the end of introduction.

Thank you for your comment, this comment is addressed by adding additional lines – 99 to 104. It summarizes the important contribution of this research to the body of knowledge.

  1. In the end of introduction, the main research questions (sub-objectives) should be mentioned.

This comment is addressed in lines –78-87 and 99-104.

  1. Please highlight the most important originalities of the research in the end introduction.

This comment is addressed in lines – 78-104.

  1. This study has been done in The United States. However, I recommend the respected authors to highlight the global value of this research in the end of introduction section. Can the results of this study be applied by the users from other parts of the world?

Thank you for your feedback, we have addressed this comment in lines – 147-153 and 306-317

  1. This paper has not the background. I am sure that there are some relative background studies around the world. Please review the literature in a separate section entitled “Background of the study”.

This comment is addressed in lines – 82-98.

  1. Methods section is very short and naïve. Please elaborate this section using adding some sub-sections explaining:

-          Research design – This is addressed in lines – 105-145.

-          Population and sampling approach – This is addressed in lines 147 to 155.

-          Research tool reliability and validity – This is addressed in lines – 105-145.

-          Measures– This is addressed in lines 147 to 155 and 105 to 124.

-          Data analysis– This is addressed in lines 146 to 155.

  1. Results section has been written and articulated very well. There is no need for further revisions.

Thank you for your comment!

  1. In discussion section please try to put your results in an international scope and then provide the readers with some useful global level recommendations. Also, in discussion section the respected authors should try compare their results with the results of other researchers in the USA and other parts of the world.

We have addressed this comment in lines – 147-153 and 306-317

  1. In conclusion section, I recommend the respected authors to mention the main take-home message of the research in a short paragraph.

We have addressed this comment in lines 342-370.

  1. Please highlight the main limitation of your study and try to draw some future pathways for the future researchers.

The limitations and the future scope of this study are mentioned in lines 364-370.

  1. In conclusion section try to highlight the main contribution of your paper to the theory and practice.

We have addressed this by rewriting the conclusion section and changes can be found in lines 342-370.

In general, I believe that this manuscript can be accepted for publication in Sustainability after MAJOR revisions.

Round 2

Reviewer 2 Report

I appreciate your responses,  which improve your manuscript.

Reviewer 3 Report

Dear authors,

Thank you very much for your efforts to improve the manuscript. I believe that the manuscript can be accepted for publication in present form. 

Best,

Reviewer